# Hybrid Resistance and Virulence Plasmids in “High-Risk” Clones of *Klebsiella pneumoniae*, Including Those Carrying *bla*_NDM-5_

**DOI:** 10.3390/microorganisms7090326

**Published:** 2019-09-06

**Authors:** Jane Turton, Frances Davies, Jack Turton, Claire Perry, Zoë Payne, Rachel Pike

**Affiliations:** 1National Infection Service, Public Health England, 61, Colindale Avenue, London NW9 5EQ, UK (C.P.) (Z.P.) (R.P.); 2Imperial College Healthcare NHS Trust, North West London Pathology, Hammersmith Hospital, Du Cane Road, London W12 0HS, UK; 3Independent Informatician, NW9 0TA c/o National Infection Service, Public Health England, 61, Colindale Avenue, London NW9 5EQ, UK

**Keywords:** virulence plasmid, *rmpA*/*rmpA2*, carbapenemase, high-risk clone, *Klebsiella pneumoniae*, nanopore sequencing

## Abstract

Virulence plasmids are associated with hypervirulent types of *Klebsiella pneumoniae*, which generally do not carry antibiotic resistance genes. In contrast, nosocomial isolates are often associated with resistance, but rarely with virulence plasmids. Here, we describe virulence plasmids in nosocomial isolates of “high-risk” clones of sequence types (STs) 15, 48, 101, 147 and 383 carrying carbapenemase genes. The whole genome sequences were determined by long-read nanopore sequencing. The 12 isolates all contained hybrid plasmids containing both resistance and virulence genes. All carried *rmpA*/*rmpA2* and the aerobactin cluster, with the virulence plasmids of two of three representatives of ST383 carrying *bla*_NDM-5_ and seventeen other resistance genes. Representatives of ST48 and ST15 had virulence plasmid-associated genes distributed between two plasmids, both containing antibiotic resistance genes. Representatives of ST101 were remarkable in all sharing virulence plasmids in which *iucC* and *terAWXYZ* were missing and *iucB* and *iucD* truncated. The combination of resistance and virulence in plasmids of high-risk clones is extremely worrying. Virulence plasmids were often notably consistent within a lineage, even in the absence of epidemiological links, suggesting they are not moving between types. However, there was a common segment containing multiple resistance genes in virulence plasmids of representatives of both STs 48 and 383.

## 1. Introduction

*Klebsiella pneumoniae* poses a dual threat to human health; in the community, it is associated with invasive disease, which can affect young, otherwise healthy people, while in the nosocomial setting it has a marked propensity to acquire antibiotic resistance [1,2,3]. It is unusual for isolates to display both these properties, but, increasingly, resistant and virulent isolates have been reported [4,5,6]. Traditionally, “hypervirulence” has been associated only with particular types, such as capsular type K1 sequence type (ST) 23 (K1-ST23) and capsular type K2 ST86 (K2-ST86), which were generally susceptible to antibiotics, but recently, virulence elements found in these types have also been described in other STs that are strongly associated with resistance, such as STs 11, 15 and 383 [7,8,9,10]. Many of these elements are carried on a large virulence plasmid, which typically carries acquired siderophore genes (especially for aerobactin, but also for salmochelin and enterochelin), heavy metal resistance genes (coding for tellurite, silver, copper and lead resistance) and capsule up-regulation genes, *rmpA* and *rmpA2*. We have previously described a representative of ST147 carrying *bla*_NDM-1_ which carried a hybrid virulence plasmid that not only contained the aerobactin cluster, *rmpA*, *rmpA2*, and tellurite resistance genes, but also the antibiotic resistance genes *sul1*, *sul2*, *armA*, *dfrA*5, *mph*(A), and *aph(3′)-VIb* (CM007852) [9]. Similarly, Lam et al. [11] have described representatives of ST15 carrying both virulence and resistance genes in a single plasmid, and a virulence plasmid carrying *bla*_KPC-2_ was reported in K1 hypervirulent *Klebsiella pneumoniae* [12]. In such cases, selection by relevant antibiotics will also select for virulence characteristics. Sequencing of further carbapenemase-gene-positive nosocomial isolates has identified twelve more isolates carrying hybrid resistance/virulence plasmids, three carrying a plasmid containing both *bla*_NDM-5_ and virulence genes, which we describe here.

## 2. Results

KpvST101_OXA-48 was isolated in June 2018 from blood from an elderly patient with septicaemia in a hospital in South East England. Typing by VNTR analysis gave a profile of 5,4,1,1,-,2,4,4,3,2,3, corresponding to ST101, confirmed from the whole genome sequence. A *bla*_OXA-48-like_ gene was detected by PCR, as were *rmpA* and *rmpA2*. The minION sequencing gave a number of fully circularised contigs corresponding to the chromosome (5453528 bp) (CP031368), the IncL/M OXA-48 plasmid (65380 bp) (CP031374) and the virulence plasmid (292699 bp) (CP031369), among others. The IncFII(K)/IncFIB(K) virulence plasmid contained *rmpA*, *rmpA2*, the aerobactin cluster (*iutA*, *iucABD*, although both *iucB* and *iucD* were truncated), copper (*pcoABCDERS*), silver (*silCERS*) and tellurite (*terBCDE*) resistance genes, other genes associated with hypervirulence (notably *cobW*, involved in cobalamin biosynthesis, *luxR*, coding for a quorum-sensing regulator controlling virulence gene expression, *pagO*, which protects bacteria from phagocytosis, *shiF*, promoting transfer of lysine) and antibiotic resistance genes *aph(6)-Id* (conferring resistance to streptomycin), *bla*_TEM-1B_ (beta-lactam resistance), *mph*(A) (macrolide resistance), *sul1*, *sul2* (sulphonamide resistance) and *dfrA5* (trimethoprim resistance) (Table 1, Figure 1a). Interestingly, this isolate carried *pld1*, coding for a phospholipase D family protein, which has been linked to hypervirulence [13], as well as the yersiniabactin cluster (*irp1*, *irp2*, *ybtAEPQSTUX*) in the chromosome, which also carried genes encoding resistance to beta-lactam antibiotics, fluoroquinolones and fosfomycin. It was resistant to most antibiotics tested (including amikacin, gentamicin, tobramycin, ampicillin, aztreonam, cefepime, cefotaxime, ceftazidime, ertapenem, piperacillin/tazobactam, temocillin, ceftolozane/tazobactam, ciprofloxacin and minocycline), with the exception of imipenem, colistin and ceftazidime/avibactam; there was reduced susceptibility to meropenem (MIC 8 mg/L) (Appendix A). All the resistance genes detected are listed in Table 1. Two other isolates of *rmpA*/*rmpA2*-positive ST101 were sequenced, one isolated in May 2018 from a further patient from the same hospital as KpvST101_OXA-48 (Kpv_ST101_SE2_2), and the other (Kpv_ST101_L5) isolated in October 2018 from a patient in a London hospital with no known epidemiological links to the other two. Remarkably, all three isolates had virulence plasmids lacking *iucC* and *terAWXYZ* (but containing *terBCDE*) and carrying truncated *iucB* and *iucD*; these are the only isolates in which this has been observed (Figure 1a–c). There were fewer resistance genes in the virulence plasmid of the London isolate (*mph(A)*, *sul1*, *dfrA5* only) than in the other two ST101 isolates, which both carried the same complement of additional genes (Table 1). All three isolates had additional plasmids carrying *bla*_OXA-48_.

KpvST383_NDM_OXA-48 was isolated in April 2018 from blood from a middle-aged patient with bacteraemia in a hospital in London. The patient suffered overwhelming sepsis and multi-organ failure, and subsequently died. This isolate had a VNTR profile of 3,1,5,2,0,2,3,3,4,2,3, corresponding to ST383, confirmed from the whole genome sequence. It was PCR-positive for *rmpA*, *rmpA2*, *bla*_NDM-like_ and *bla*_OXA-48-like_. Sequencing gave several circular contigs, corresponding to the chromosome (5361371 bp) (CP034200), the OXA-48 plasmid (72057 bp) (CP034202), and a hybrid NDM/virulence plasmid (372826 bp) (CP034201). This plasmid contained the aerobactin cluster (*iutA*, *iucABCD*), *rmpA*/*rmpA2*, tellurite resistance genes *terABCDEWXYZ*, *cobW*, *luxR*, *pagO*, and *shiF* and resistance genes *bla*_NDM-5_, *bla*_CTXM-15_, *bla*_OXA-9_, *qnrS*1, *bla*_TEM-1B_, *dfrA5*, *sul1*, *sul2*, *armA*, *aph(3′)-Ia*, *aph(3′)-VI*, *aac(6′)-lb*, *aadA1*, *aac(6′)-lb-cr*, *mph*(A), *mph*(E), *msr(E)* and *catA1* (all at 100% identity, with the exception of *catA1* (99.11% identity)), with the resistance genes located in two distinct areas (Figure 1d). Genes conferring tetracycline resistance (*tet*(A)), fosfomycin resistance (*fosA*), quinolone resistance (*oqxA*, *oqxB*) and further beta-lactamase (*bla*_SHV-26_) genes were located on the chromosome (Table 1). The isolate did not carry the yersiniabactin cluster, often found in an integrative conjugative element (ICE) in the chromosome. In common with KpvST101_OXA-48, it carried *pld1* and *fyuA* on the chromosome. The virulence plasmid is highly similar to that of KpvST147L_NDM (CM007852) [9], which has 99% identity with it, but only 93% coverage, this mainly being accounted for by the stretch from approximately 60,000 to 95,000 bp in this isolate, which contains *bla*_NDM-5_, *aph(3′)-VI*, *qnrS*1, *bla*_CTX-M-15_, *aac(6′)-Ib*, *aac(6’)-Ib-cr*, *aadA1*, *bla*_OXA-9_ and *bla*_TEM-1B_, missing in KpvST147L_NDM. There is an IS*26*-like element (also referred to as IS*6* family transposase) at nucleotides (nt) 66471 to 67174, and a partial copy at 90579 to 90949, among other instances of this and other insertion sequences and transposons in the sequence (Figure 2). Two further *rmpA*/*rmpA2*-positive isolates of ST383 were sequenced, separated in space and time from KpvST383_NDM_OXA-48 (Kpv_ST383_L2, isolated in 2017 in a different London hospital, and Kpv_ST383_S1, isolated in 2016 in a Scottish hospital) (Figure 1e,f). All carried the same complement of virulence/metal resistance genes, with the exception of Kpv_ST383_L2, which lacked *terXY*, but, in common with the others, carried the rest of the tellurite resistance cluster (*terABCDEWZ*). All three lacked the yersiniabactin cluster (*irp1*, *irp2*, *ybtAEPQSTUX*). Kpv_ST383_NDM_OXA-48 and the Scottish isolate shared the same 18 resistance genes in the virulence plasmid, including *bla*_NDM-5_, while the second London isolate carried only *aph(3’)-Ia*, *aadA1*, *sul1* and *tet(B)* in the virulence plasmid, the latter not found in the virulence plasmids of the other two. In contrast to the others, it carried *bla*_NDM-1_ in a separate plasmid. All three carried *bla*_OXA-48_ in a further plasmid. Essentially, KpvST383_NDM_OXA-48 and the Scottish isolate shared the same hybrid resistance/virulence plasmid, the latter sharing 99% coverage and 98.8% identity with the former (CP034201), despite being isolated over 2 years and hundreds of miles apart and with no known links between them. However, the virulence plasmid of Kpv_ST383_L2 shared only 74% coverage with that of Kpv_ST383_NDM_OXA-48. Both Kpv_ST383_L2 and Kpv_ST383_S1 were resistant to most antibiotics tested, including ceftazidime/avibactam; Kpv_ST383_L2 was susceptible to tigecycline (MIC 0.5 mg/L) and fosfomycin (MIC 32 mg/L), but resistant to colistin (MIC 32 mg/L), while Kpv_ST383_S1 was susceptible to colistin (MIC 1 mg/L) and fosfomycin (MIC 16 mg/L), but resistant to tigecycline (MIC 4 mg/L) (Appendix A).

Three representatives of ST147 carrying virulence plasmids (Kpv_ST147B_SE1_1_NDM, Kpv_ST147_SE1_2, Kpv_ST147_L3) were sequenced (Table 1); in all three cases, the virulence plasmids were a similar size and carried the same virulence gene content and very similar resistance gene complements as each other and the previously described KpvST147L_NDM (CM007852) [9], despite being isolated in three different hospitals and over a three-year time span (Table 1; Figure 1g–j). There were potential epidemiological links between the three isolates described here, especially those from the same hospital, but not with the original isolate from 2016. All three isolates carried *bla*_NDM-1_ in a separate IncFIB(pQil) plasmid of approximately 54 kb (e.g., CP040728), which was highly similar between the three, supporting that they may be linked (all share 99% coverage and < 99% identity with one another; they also share similarly high homology with other NDM plasmid sequences on GenBank (e.g., AP018834); Single Nucleotide Polymorphism (SNP) analysis between the Illumina-corrected chromosomal sequences of Kpv_ST147B_SE1_1_NDM and Kpv_ST147_SE1_2 also supported this (six SNPs difference). Kpv_ST147_SE1_2 and KpvST147L_NDM were susceptible only to colistin (MICs ≤ 0.5 and 1 mg/L, respectively) (Appendix A).

Kpv_ST15_NDM and Kpv_ST15_NW1 were isolated in December 2016 and September 2018 from the throat and urine of patients in hospitals in London and North West England, respectively. These representatives of ST15 displayed some considerable differences in their virulence plasmids; while both contained the aerobactin cluster (*iutA*, *iucABCD*), *rmpA* and *rmpA2*, the latter carried the tellurite resistance cluster (*terABCDEWXYZ*), *pagO* and *luxR,* not found in the former (Figure 1k,l). These are genes that are generally found in virulence plasmids, so their absence is notable in Kpv_ST15_NDM. Moreover, Kpv_ST15_NW1 carried the copper (*pcoABCDERS*) and silver (*silCERS*) resistance genes in a second plasmid (also carrying *tet(D)* and *bla*_SHV-1_), while these were carried in the virulence plasmid of Kpv_ST15_NDM (CP040595). The virulence plasmid of Kpv_ST15_NDM carried lead resistance genes (*pbrABCR*), not found in any of the other plasmids. Both virulence plasmids also contained a number of antibiotic resistance genes, but they were different in the two, *sul1* being the only common one between them. Both isolates carried *bla*_NDM-1_ in a separate plasmid. Both carried the yersiniabactin cluster in the chromosome, but Kpv_ST15_NDM lacked some of the *mrk* fimbrial genes that are commonly found in *K. pneumoniae* and are associated with virulence. Susceptibility testing was carried out on Kpv_ST15_NDM, which was resistant to most antibiotics tested, including tigecycline and ceftazidime/avibactam, with the exception of colistin (MIC ≤ 0.5 mg/L) and fosfomycin (MIC 16mg/L) (Appendix A).

The only carbapenemase-gene-positive isolate of ST48 that we have noted carrying a virulence plasmid (KpvST48_NDM) was from a patient from the same London hospital group (L5) as three other isolates in this study and was isolated in October 2018 from a rectal screen. Surprisingly, it also had the virulence and metal resistance elements associated with virulence plasmids distributed between two different plasmids. The main virulence plasmid of 302 kb (pKpvST48_1, CM016731), of type IncFIB(Mar), contained the aerobactin cluster (*iutA*, *iucABCD*), *rmpA*, *rmpA2* and other virulence plasmid-associated genes (e.g., *cobW*, *luxR*, *pagO*, *shiF*) (Figure 1m), while the second plasmid of 226.8 kb (pKpvST48_2, CM016732) carried the lead and silver resistance gene clusters (*pcoABCDERS and silCERS*). Both plasmids also carried antibiotic resistance genes (*aph(3’)-Ia aph(3’)-VI aac(6’)-Ib aadA1*, *bla*_NDM-5_, *bla*_CTX-M-15_, *bla*_OXA-9_, *bla*_TEM-1B_, *qnrS1*, *aac(6’)-Ib-cr*, *catA* in the main virulence plasmid, and *tet(D)* in the second one). This is therefore a further example of a virulence plasmid carrying *bla*_NDM-5_ (in a different plasmid to those found in KpvST383_NDM_OXA-48/Kpv_ST383_S1); remarkably, the whole segment of pKpvST383L (CP034201 from KpvST383_NDM_OXA-48) containing antibiotic resistance genes including *bla*_NDM-5_ detailed in Figure 2 is duplicated in pKpvST48_1 (CM016731); indeed, this plasmid had 84% coverage of pKpvST383L, despite being some 70 kb smaller, including a large segment (nt 106473-283838) corresponding to nt 42544 to 220502 of pKpvST383L (CP034201), which carried the resistance genes above and ended with an IS110 family transposase.

The virulence plasmids from the representatives of “high-risk” clones were compared with one another, and with pK2044 (AP006726), the virulence plasmid from a representative of K1-ST23 (NTUH-K2044) of the KpVP-1-dominant virulence plasmid type [15] (Figure 1). This showed that all the plasmids shared virulence regions with pK2044, but that those from representatives of ST101 were quite considerably different from those of representatives of STs 147 and 383, which were very similar to one another, other than obvious differences such as the extra segment (containing *bla*_NDM-5_) in pKpvST383L (CP034201) and pKpvST383_S1 (from Kpv_ST383_S1) (Figure 1a,b) (also in pKpvST48_1). pKpvST15 (CP040595 from Kpv_ST15_NDM) was different from all the other plasmids. All the plasmids carried genes coding for plasmid transfer and various conjugal transfer proteins such as TraABDEFGIJNPUVY, suggesting that conjugative plasmid transfer is possible. Our study suggests that transfer of complete plasmids is unlikely, but that IS/transposon-bound segments of sequence containing antibiotic resistance genes may move.

In at least three of the cases described here, the isolates were associated with sepsis, supporting that they may be associated with hypervirulence. A number of others were from screening swabs; this is not a contra-indication to hypervirulence, since it is recognised that colonisation precedes infection in recognised hypervirulent types. Fortunately, the incidence of high-risk clones with virulence plasmids was still relatively low, even among submissions to the reference laboratory, which are enhanced both for carbapenemase producers and for potentially hypervirulent isolates; we have noted them since 2016, and have found a total of 40 nonduplicate isolates among some 3,600 isolates, screened up until the end of the first quarter of 2019.

## 3. Discussion

The high-risk clones described here (of STs 15, 48, 101, 147 and 383) are not simply gaining the virulence plasmids found in hypervirulent types, which do not generally contain antibiotic resistance genes, but carry their own versions that combine both resistance and virulence elements. The only example where a virulence plasmid in a classical hypervirulent type has been described with a resistance gene at the time of writing is the hybrid KPC-2 virulence plasmid in a hypervirulent isolate of K1-ST23, which has a conserved virulence plasmid similar to those found in other isolates of K1-ST23, but with an antibiotic resistance region flanked by two copies of IS*26* [12]. The authors concluded that multiple insertion elements (IS) were responsible for mediating the plasmid recombination. Analysis by Lam et al. [16] of the virulence/resistance plasmid in our previously described representative of the high-risk clone ST147, KpvST147L_NDM, showed that it carried sequences from KpVP-1, (the dominant virulence plasmid type found in K1-ST23 hypervirulent isolates such as NTUH K2044 (AP006726.1)) (40% coverage), an IncFII *tra-trb* conjugative transfer region and transposons carrying antibiotic resistance genes. The presence of multiple transposons and insertion sequences was a consistent finding for all the isolates described here; for example, there are seven (plus a truncated version) copies of IS*26*-like insertion sequences in pKpvST383, six in pKpvST147L, and four in pKpvST101 (as defined by comparison with nucleotides 1–717 of KU984333). It therefore seems likely that these hybrid plasmids have been created by IS-mediated recombination. It is important to recognise the existence of hybrid virulence and resistance plasmids, so that an antibiotic is not chosen inadvertently that selects for isolates carrying virulence plasmids. The combination of an NDM gene and virulence elements in the same plasmid, as seen in KpvST383_NDM_OXA-48, Kpv_ST383_S1 and KpvST48_NDM is particularly alarming. In several cases described here, the clinical outcome was clearly consistent with enhanced virulence, in contrast to that described previously where the isolate (KpvST147L_NDM) was from a rectal screen. Another study describing carbapenemase-gene-positive representatives of ST15 carrying a virulence plasmid (of approximately 178 kb and carrying *rmpA2* and the *iutAiucABCD* aerobactin gene cluster) found that the isolates were not hypervirulent based on assays looking for survival in neutrophils and killing of *Galleria mellonella* and that all the affected patients recovered [17]. The authors concluded that further studies were needed to understand the relationship between the hypervirulent phenotype, carriage of the pLVPK-like virulence plasmid and capsular type. However, representatives of ST11 carrying a virulence plasmid showed enhanced survival in neutrophils and killed 100% of *G. mellonella*; these isolates were responsible for fatal infections, and the authors concluded that the hypervirulence was due to acquisition of a 170 kb pLVPK-like virulence plasmid by classic ST11 carbapenem-resistant *K. pneumoniae* strains [8]. Another study links three glycosyltransferases encoded by the cps gene cluster, GT-1, GT-2 and WcaJ, with virulence [18]. We note the presence of *wcaJ* (and glycosyltransferases) in the cps gene cluster of KpvST383_NDM_OXA-48 and KpvST147B_SE1_1_NDM, but not in those of KpvST48_NDM, Kpv_ST15_NDM or KpvST101_OXA-48.

While some of the hybrid resistance/virulence plasmids described here were similar, each ST was nevertheless associated with a different plasmid, suggesting that there is an IS-mediated mechanism for combining these genes in these plasmids. In one instance there was clearly a shared antibiotic resistance region in the virulence/resistance plasmids of isolates of different types (STs 383 and ST48), again supporting IS-mediated recombination. The existence of hybrid virulence and resistance plasmids requires enhanced surveillance, especially since it is occurring in high-risk clones that are already adept at acquiring resistance genes, if potentially untreatable invasive infections are to be avoided. 

## 4. Materials and Methods 

Isolates were submitted to the laboratory from hospitals in the United Kingdom for typing and cross-infection investigation, and/or susceptibility determination. They were identified as potentially hypervirulent by detection of *rmpA* and/or *rmpA2* in a multiplex PCR seeking capsular types K1, K2, K5, K20, K54 and K57, *rmpA*, *rmpA2* and *wcaG*, among other targets [9]. Typing was by Variable Number Tandem Repeat (VNTR) analysis at 11 loci [9], while detection of carbapenemase genes was carried out in-house by the submitting hospital. Representative isolates that were PCR-positive for *rmpA*/*rmpA2*, belonged to high-risk clones (identified by VNTR typing) and in which a carbapenemase gene(s) had been detected were selected for this study. Isolates were sequenced on a minION Mk1B, following DNA extraction using a GeneJet genomic DNA kit (ThermoFisher, Loughborough, United Kingdom) and library preparation using a Rapid Barcoding Kit SQK-RBK004 (Oxford Nanopore Technologies, Oxford, United Kingdom). The barcoded libraries of up to nine isolates were pooled, concentrated using Ampure XP beads (Beckman Coulter, High Wycombe, United Kingdom) and run on a FLO-MIN106 (R9 version) flowcell using minKnow software with local basecalling. Barcodes were identified and removed using Guppy barcode software available from Oxford Nanopore technologies and reads for each barcode assembled using miniasm 0.2-r168-dirty [19], followed by three iterations of correction with Racon 1.3.1 [20]. Illumina sequencing was also carried out on six isolates, as described previously, and the short-read sequences used to correct the minION assemblies using ten iterations of Racon. Sequences were submitted to GenBank under BioProject PRJNA483246 and BioSamples SAMN09729079 (KpvST101_OXA-48) (accession numbers CP031368-CP031375), SAMN10409888 (KpvST383_NDM_OXA-48) (accession numbers CP034200-CP034202), SAMN11793480 (KpvST48_NDM) (accession numbers VCEE00000000; CM016731-CM016736), SAMN11793565 (KpvST15_NDM) (accession numbers CP040593-CP04599) and SAMN11835544 (KpvST147B_SE1_1_NDM) (accession numbers CP040724-CP040731). The minION fastq files of all the remaining isolates (and the Illumina fastqs, where available) were submitted to the Sequence Read Archive (SRA) accession numbers SAMN12212724 (KpvST101_SE2_2), SAMN12216748 (Kpv_ST101_L5), SAMN12216749 (Kpv_ST147_SE1_2), SAMN12216750 (Kpv_ST147_L3), SAMN12221326 (Kpv_ST383_L2), SAMN12221327 (Kpv_ST383_S1) and SAMN12221328 (Kpv_ST15_NW1). Although greater than 99% accurate, the minION sequences have some additional/missing bases and therefore are not completely accurate, but have the huge advantage of facilitating complete assemblies. Virulence plasmids were compared with that of a previously described isolate of ST147 (KpvST147L_NDM) (CM007852) and with that in NTUH-K2044, a representative of “hypervirulent” K1-ST23 (AP006726). Sizes of plasmids were confirmed by pulsed-field gel electrophoresis (PFGE), as described by Barton et al. [21], following digestion of agarose-embedded DNA plugs in 100 µL 1 × S1 nuclease buffer containing 0.1 U/µL S1 nuclease (Thermofisher, Loughborough, United Kingdom) (Appendix A). PFGE conditions were 6 V/cm for 30 h at 12 °C, with initial and final switch times of 1 s and 50 s. 

MICs were determined by agar dilution and interpreted using EUCAST breakpoints [15]. Resistance genes and plasmid replicon types were detected using ResFinder and PlasmidFinder, respectively, on the Center for Genomic Epidemiology website [22,23]. Sequence types were determined from the Pasteur Institute *Klebsiella pneumoniae* MLST website (available online: https://bigsdb.pasteur.fr/cgi-bin/bigsdb/bigsdb.pl?db=pubmlst_klebsiella_seqdef&page=sequenceQuery).

## Figures and Tables

**Figure 1 microorganisms-07-00326-f001:**
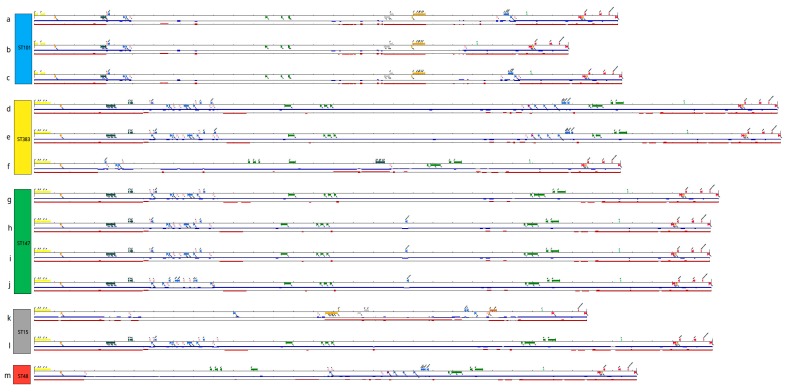
Comparison of the virulence plasmids found in (**a**) KpvST101_OXA-48 (pKpvST101, CP031369), (**b**) Kpv_ST101_L5, (**c**) Kpv_ST101_SE2_2, (**d**) KpvST383_NDM_OXA-48 (pKpvST383L, CP034201), (**e**) Kpv_ST383_S1 (pKpvST383_S1), (**f**) Kpv_ST383_L2, (**g**) KpvST147L_NDM (pKpvST147L, CM007852), (**h**) Kpv_ST147B_SE1_1_NDM (pKpvST147B, CP040726), (**i**) Kpv_ST147_SE1_2, (**j**) Kpv_ST147_L3, (**k**) Kpv_ST15_NDM (pKpvST15, CP040595), (**l**) Kpv_ST15_NW1, (**m**) Kpv_ST48_NDM ((pKpvST48_1, CM016731). Sequences are shown starting from *iucA* in the aerobactin cluster, with aerobactin genes (*iutA*, *iucABCD*) coloured yellow, *rmpA* and *rmpA2* coloured orange, other virulence gene-associated genes (*cobW*, *hemin*, lysozyme inhibitor, *shiF*, *luxR*, *pagO*, SAM-dependent methyltransferase (SAM-dmt)) crimson, *bla*_NDM-5_ purple, other antibiotic resistance genes royal blue, tellurite resistance genes (*terABCDEWXYZ*) in dark grey, copper resistance genes (*pcoABCDERS*) in gold, silver resistance genes (*silCERS*) in silver, lead resistance genes (*pbrABCR*) in brown and genes encoding various Tra conjugal transfer proteins in green. IS26-like elements are marked in pink. Colour blocks above the lines indicate forward sequences, while those below the line show sequences in the opposite orientation. The red line under each sequence shows BLAST alignments with pK2044 (the virulence plasmid from NTUH2044, AP006726). The dark blue line shows BLAST alignments with pKpvST147L (CM007852 from KpvST147_NDM). Kpv_ST15_NW1 and Kpv_ST48_NDM both carried a further plasmid with the *pcoABCDERS* and *silCERS* genes.

**Figure 2 microorganisms-07-00326-f002:**
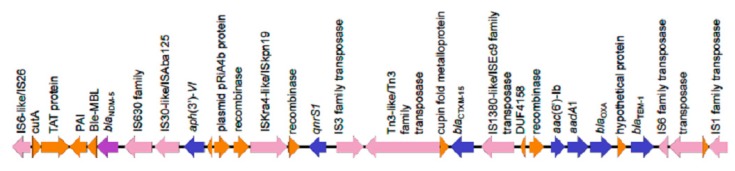
Genes encoded in the region of nt 66471 to 93154 of CP034201, the virulence plasmid of KpvST383_NDM_OXA-48, which includes *bla*_NDM-5_ (coloured purple). Other antibiotic resistance genes are shown in blue, while transposons are shown in pink. This same sequence of genes was also found in pKpvST48_1 of Kpv_ST48_NDM (nt 130384 to 156980 of CM016731). Abbreviations are: nt, nucleotide; TAT, twin-arginine translocation pathway signal sequence domain; Ble-MBL, bleomycin-binding protein. Figure created using Easyfig software [14].

**Table 1 microorganisms-07-00326-t001:** Summary of virulence and resistance gene characteristics of thirteen representatives of high-risk clones carrying hybrid virulence/resistance plasmids. All isolates were subjected to nanopore sequencing; six isolates were also subjected to Illumina sequencing, and the Illumina-corrected sequences are described below. Plasmids are described by their GenBank accession numbers or the “utg” contig number from the assembly, ‘c’ after the number indicates a circular contig.

Isolate	Virulence Plasmid	Other Virulence Factors (Chromosome)	Antibiotic Genes in Virulence Plasmid	Other Resistance Genes	Hospital (Code), Date and Source of Isolation	Clinical Manifestation	Comments
KpvST101_OXA-48 Illumina corrected sequence	plasmid pKpvST101 CP031369 292699 bp *rmpA*, *rmpA2*, (*iutA*, *iucABD*, (*iucB* and *iucD* truncated), *pcoABCDERS*, *silCERS terBCDE*, *cobW*, *luxR*, *pagO*, *shiF*	*fyuA*, *kfuABC*, *irp1*, *irp2*, *ybtAEPQSTUX*, *mrkABCDFHIJ*, *pld1*	*aph(6)-Id*, *aph(3’)-Ib*, *bla*_TEM-1B_*mph(A)*, *sul1*, *sul2* and *dfrA5*	Chromosome *blaSHV-28*, *bla*_CTX-M-15_, *oqxA*, *oqxB*, *fosA* plasmid CP031374 65380 bp *bla*_OXA-48_ plasmid CP031372 210661 bp *armA*, *msr(E)*, *mph(E)* plasmid CP031373 43670 bp *blaSHV-28*, *qnrS1*	South_East_2 (SE2) June 2018 blood	septicaemia	no *iucC* no *terAWXYZ*
Kpv_ST101_SE2_2	utg000004c (294751 bp) *iutA*, *iucABD* (*iucB* and *iucD* truncated), *rmpA*, *rmpA2*, *terBCDE*, *pcoABCDERS*, *silCERS*, *FecA*, *cobW*, *luxR*, *pagO*, *shiF*	*fyuA*, *kfuABC*, *irp1*, *irp2*, *ybtAEPQSTUX*, *pld1*, *mrkABCDFHIJ*	*aph(6)-Id*, *aph(3’)-Ib*, *bla*_TEM-1B_, *mph(A)*, *sul1*, *sul2* and *dfrA5*	Chromosome *blaSHV-28*, *bla*_CTX-M-15_, *oqxA*, *oqxB*, *fosA*, utg16c; 65307 bp *bla*_OXA-48_ utg31c *bla*_CTX-M-15_, *armA*, *msr(E)*, *mph(E)* utg27c *blaSHV-28*, *qnrS1*	South_East_2 (SE2) May 2018 sputum	chest infection	no *iucC* no *terAWXYZ*
Kpv_ST101_L5	utg00035c (267754 bp) *iutA*, *iucABD* (*iucB* and *D* truncated), *rmpA*, *rmpA2*, *terBCDE*, *pcoABCDERS*, *silCERS*, *FecA*, *cobW*, *luxR*, *pagO*, *shiF*	*fyuA*, *irp1*, *irp2*, *kfuABC*, *ybtAEPQSTUX*, *pld1*, *mrkABCDFHIJ*	*mph(A)*, *sul1*, *dfrA5*	Chromosome *blaSHV-28*, *bla*_CTX-M-15_, *oqxA*, *oqxB*, *fosA* utg000020c *armA*, *msr(E)*, *mph(E)* utg000037c *bla*_OXA-48_ utg000108c *blaSHV-28*, *qnrS1*	London_5 (L5) October 2018 rectal screen	screen	no *iucC* no *terAWXYZ*
KpvST383_NDM_OXA-48 Illumina corrected sequence	plasmid pKpvST383L CP034201 372826 bp *iutA*, *iucABCD*, *rmpA*/*rmpA2*, *terABCDEWXYZ*, *cobW*, *luxR*, *pagO*, *shiF*	*mrkABCDFHIJ*, *pld1*	*bla*_NDM-5_, *bla*_CTXM-15_, *bla*_OXA-9_, *qnrS*1, *bla*_TEM-1B_, *dfrA5*, *catA1*, *sul1*, *sul2*, *armA*, *aph(3′)-1a*, *aph(3′)-VI*, *aac(6′)-lb*, *aadA1*, *aac(6′)-lb-cr*, *mph(A)*, *mph (E)* and *msr(E)*	*blaSHV-26*, *oqxA*, *oqxB*, *fosA*, *mph(A)*, *catA1*, *tet(A)* plasmid CP034202 72057 bp *aph(6)-Id*, *aph(3’)-Ib*, *aph(3’)-VIb*, *aph(3’)-Ib*, *blaCTX-M-14b*, *bla*_OXA-48_	London_5 (L5) April 2018 blood	bacteraemia, sepsis, multi-organ failure and death	no yersiniabactin
Kpv_ST383_L2	utg000003c (294,141) *iutA*, *iucABCD*, *rmpA*/*rmpA2*, *terABCDEWZ*, *cobW*, *luxR*, *pagO*, *shiF*	*mrkABCDFHIJ*, *pld1*	*aph(3’)-Ia*, *aadA1*, *sul1*, *tet(B)*	*blaSHV-26*, *oqxA*, *oqxB*, *fosA*, *sul1* utg000002c (110455 bp) *rmtC*, *bla*_NDM-1_ utg000011c (105957 bp) *aph(’’)-Ib*, *aph(6)-Id*, *aac(6’)-Ib3*, *aadA1*, *bla*_OXA-48_ *bla*_CTX-M-14b_, *bla*_OXA-9_, *aac(6’)-Ib-cr*	London_2 (L2) April 2017 blood	sepsis	Not got yersiniabactin *ybt* cluster, *irp1,2*) or *terXY*
Kpv_ST383_S1	utg000038c (374430 bp) (pKpvST383_S1) *iutA*, *iucABCD*, *rmpA*, *rmpA2*, *terABCDEWXYZ*, *cobW*, *pagO*, *shiF*, *luxR*	*mrkABCDFHIJ*, *pld1*	*aadA1*, *aac(6’)-Ib aph(3’)-Ia*, *armA*, *aph(3’)-VI*, *bla*_TEM-1B_, *bla*_OXA-9_, *bla*_CTX-M-15_, *bla*_NDM-5_, *aac(6’)-Ib-cr*, *qnrS1*, *msr(E)*, *mph(E)*, *mph(A)*, *catA1*, *sul1*, *sul2*, *dfrA5*	*blaSHV-26*, *oqxA*, *oqxB*, *fosA*, *mph(A)*, *catA1*, *tet(A)* utg000039c (72122 bp) *aph(3’)-Ib*, *aph(3’)-VIb*, *aph(3’)-Ib*, *aph(6)-Id*, *bla*_OXA-48_, *bla*_CTX-M-14b_	Scotland1 (S1) Feb 2016 rectal swab	No information; screening swab	*terXY* present, no *ybt irp1,2* cluster Ex Egypt (in hospital in Cairo)
KpvST147L_NDM previously described [9]; Illumina corrected	plasmid pKpvST147L CM007852.1 343,282 bp *iutA*, *iucABCD*, *rmpA*, *rmpA2*, *terABCDEWXYZ*, *cobW*, *luxR*, *pagO*, *shiF*	*mrkABCDFHIJ*	*sul1*, *sul2*, *armA*, *dfrA*5, *mph(A)*, *msr(E)*, *mph(E)*, *aph(3’)-Ia*	*bla*_NDM-1_, *aph(3’)-Ia*. *aac(6’)-Ib*, *aadA1 blaTEM-1A*, *blaOXA-9*, *blaCTX-M-15*, *blaSHV-67*, *qnrS1*, *aac(6’)-Ib-cr*, *oqxA*, *oqxB*, *fosA*	London_5 (L5) January 2016 rectal swab	screening swab	
KpvST147B_SE1_1_NDM Illumina corrected sequence	plasmid pKpvST147B CP040726 (339117 bp) *iutA*, *iucABCD*, *rmpA*, *rmpA2*, *terABCDEWXYZ*, *cobW*, *luxR*, *pagO*, *shiF*	*fyuA*, *mrkABCDFHIJ*, *irp1*, *irp2*, *ybtAEPQSTUX*	*sul1*, *sul2*, *armA*, *dfrA*5, *mph(A)*, *msr(E)*, *mph(E) bla*_CTX-M-15_	*aac(6’)-Ib*, *aadA1*, *blaSHV-67*, *bla*_OXA-9_, *bla*_TEM-1A_, *oqxA oqxB*, *aac(6’)-Ib-cr fosA* plasmid CP040728 53950 bp *aac(6’)-Ib-cr*, *aph(3’)-VI*, *bla*_OXA-1_, *bla*_CTX-M-15_, *bla*_NDM-1_, *qnrS1*, *catB3*, *ARR-3*, *sul1*	South_East_1 (SE1) January 2019 rectal swab	no information, screening swab	
Kpv_ST147_SE1_2 Illumina corrected sequence	utg000005c (338588 bp) *iutA*, *iucABCD*, *rmpA*, *rmpA2*, *terABCDEWXYZ*, *cobW*, *luxR*, *pagO*, *shiF*	*fyuA*, *mrkABCDFHIJ*, *irp1*, *irp2*, *ybtAEPQSTUX*	*armA*, *mphA*, *sul1*, *sul2*, *dfrA5*, *bla*_CTX-M-15_, *mph(E)*, *msr(E)*	Chromosome *oqxA*, *oqxB*, *fosA*, *blaSHV-67* utg000020c; 53949 bp, *bla*_NDM-1_, *sul1*, *aph(3’)-VI*, *catB3*, *ARR-3*, *qnrS1*, *aac(6’)-Ib-cr*, *blaOXA-1* utg000007c 39671 bp *aadA1*, *aac(6’)-Ib*, *blaTEM-1A*, *bla*_OXA-9_, *aac(6’)-Ib-cr*	South_East_1 (SE1) December 2018 urine	patient died	
Kpv_ST147_L3	utg000005c (339641 bp) *iutA*, *iucABCD*, *rmpA*, *rmpA2*, *terABCDEWXYZ*, *cobW*, *luxR*, *pagO*, *shiF*	*fyuA*, *mrkABCDFHIJ*, *irp1*, *irp2*, *ybtAEPQSTUX*	*armA*, *aph(3’)-Ia*, *bla*_CTX-M-15_, *mph(A)*, *mph(E)*, *msr(E)*, *sul1*, *sul2*, *dfrA5*	Chromosome *blaSHV-67*, *oqxA*, *oqxB*, *fosA* utg000004c; 53895 bp *bla*_NDM-1_, *aac(6’)-Ib-cr*, *aph(3’)-VI*, *bla*_OXA-1_, *bla*_CTX-M-15_, *aac(6’)-Ib-cr*, *qnrS1*, *catB3*, ARR-3, *sul1*	London_3 (L3) January 2019 not given	No information	
Kpv_ST15_NDM (L6) Illumina corrected sequence	plasmid pKpvST15 CP040595 277162 bp *iutA*, *iucABCD*, *rmpA*, *rmpA2*, *pbrABCR*, *pcoABCDERS*, *silCERS*, *cobW*, *shiF*	*fyuA*, *irp1*, *irp2*, *ybtAEPQSTUX*, *kfuABC*, *mrkA(B)*,	*aac(6’)-Ib3*, *rmtC*, *bla*_CMY-6_, *aac(6’)-Ib-cr*, *sul1*	*bla*_CTX-M-15_, *bla*SHV-28, *oqxA*, *oqxB*, *fosA*, plasmid CP040598 106597 bp *bla*_NDM-1_, *sul1*	London_6 (L6) 19.12.16 Throat swab	screen	*mrkB* truncated (17% only), no *mrkCDFHIJ*, no tellurite resistance genes, *pagO* or *luxR*
Kpv_ST15_NW1	utg000104c (340,126 bp): *iutA*, *iucABCD*, *rmpA*, *rmpA2*, *terABCDEWXYZ*, *cobW*, *luxR*, *pagO*, *shiF* 2nd plasmid: utg000031c 222308 bp: *pcoABCDERS*, *silCERS*	*irp1*, *irp2*, *mrkABCDFHIJ*, *ybtAEPQSTUX*, *kfuABC*	*armA*, *aph(3’)-Ia*, *msr(E)*, *mph(E)*, *mph(A)*, *sul1*, *sul2*, *dfrA5* utg000031c: *blaSHV-1*, *tet(D)*	*aph(3’)-VI*, *bla*_NDM-1_, *blaSHV-28*, *qnrS1* (utg000030l), *oqxA*, *oqxB* (utg000001l),	North_West_1 (NW1) September 2018 urine	No information	*pcoABCDERS*, *silCERS*, *tet(D)* in 2nd plasmid
Kpv_ST48_NDM Illumina corrected sequence	plasmid pKpvST48_1 CM016731 302,220 bp *iutA*, *iucABCD*, *rmpA*, *rmpA2*, *cobW*, *luxR*, *pagO*, *shiF* 2nd plasmid KpvST48_2 CM016732 226800 bp *pcoABCDERS*, *silCERS*, *FecA*	*mrkABCDFHIJ*, *irp1*, *irp2*, *ybtAEPQSTUX*, *fyuA*	CM016731: *aph(3’)-Ia aph(3’)-VI aac(6’)-Ib aadA1*, *bla*_NDM-5_, *bla*_CTX-M-15_, *bla*_OXA-9_, *bla*_TEM-1B_, *qnrS1*, *aac(6’)-Ib-cr*, *catA1* CM016732: *tet(D)*	*aadA2*, *aph(3’)-Ia*, *blaSHV-172*, *blaCTX-M-14b*, *oqxA*, *oqxB*, *fosA*, *mph(A)*, *sul1*, *dfrA12*	London_5 (L5) October 2018 rectal screen	No information; screening swab	*pcoABCDERS*, *silCERS*, *FecA*, *tet(D)* in CM016732, no tellurite resistance genes, patient ex Egypt

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
