# Peer review of "Hybrid Resistance and Virulence Plasmids in “High-Risk” Clones of Klebsiella pneumoniae, Including Those Carrying blaNDM-5"

_microorganisms, 2019, doi:10.3390/microorganisms7090326_

Round 1

Reviewer 1 Report

General comment:

The authors performed the nucleotide sequencing of virulence plasmids found in 12 K. pneumoniae isolates carrying carbapenemase genes and belonging to high-risk clones. All isolates carried antimicrobial resistance genes, including blaNDM-5 in three isolates, on virulence plasmids. Although virulence plasmids of the isolates belonging to a specific clone had a certain level of similarity, those of isolates belonging to different clones had significant difference in genetic structures. Nevertheless, surrounding genetic environment of blaNDM-5 was common among two ST383 isolates and a ST 48 isolate. The Presentation of the methods and results is clear in general. In my opinion, the scientific content of the manuscript is of interest to many readers of the journal. I have only a few minor suggestions.

Minor comments:

Materials and Methods: The process and criteria, with which the isolates had been submitted to the laboratory from each health care facility should be explained briefly. Results: Results of drug susceptibility testing should be shown separately in a new table. Line 66: According to the information in Table 1, it appears that Illumina sequencing was performed in five (not six) isolates in this study. Line 224-252: The names of the plasmids appearing in this section (e.g., pKpvST383) should be defined in the text here or beforehand. The names of the plasmids should also be provided in Table 1 and Figure 1. Figure 1: I cannot read the letters (maybe) in the figure due to the low resolution of the figure. Some modifications are required.

Reviewer 2 Report

This draft is interesting and have merit for publication

Overall, this is a clear, concise, and well-written manuscript.

The introduction is relevant and theory based. Sufficient information about the previous study findings is presented for readers to follow the present study rationale and procedures.

The methods are generally appropriate .Overall, the results are clear and compelling with two possible exceptions as below

1 The conclusion may be more concise.

2 The author may draw a flow-chart for the procedure of this experiment.

Author Response

Response to Reviewer 2:

Thank you to the reviewer for their time and trouble in reading the manuscript and for their positive and helpful comments.

Reviewer 2

This draft is interesting and have merit for publication

Overall, this is a clear, concise, and well-written manuscript.

The introduction is relevant and theory based. Sufficient information about the previous study findings is presented for readers to follow the present study rationale and procedures.

The methods are generally appropriate .Overall, the results are clear and compelling with two possible exceptions as below

Thank you so much for your encouraging and positive comments.

1 The conclusion may be more concise.

Response: The concluding paragraph of the Discussion is quite brief (8 lines) and we would find it difficult to make it more concise. But some parts of the Discussion are maybe a little repetitive and we have therefore omitted the sentence ‘Searching the plasmid sequences using IS finder showed partial matches with a number of other insertion sequence elements, especially of Tn3 family transposases’ since it does not really add anything and re-numbered the subsequent references accordingly.

2 The author may draw a flow-chart for the procedure of this experiment.

Response: Basically, representative isolates going through the laboratory routine that were PCR positive for rmpA/rmpA2, did not belong to recognised hypervirulent types such as K1-ST23 or K2-ST86, and in which a carbapenemase gene(s) had been detected were subjected to nanopore sequencing and their virulence plasmids compared. In response to reviewer 1 we have added ‘Isolates were submitted to the laboratory from hospitals in the United Kingdom for typing and cross-infection investigation, and/or susceptibility determination’ at the beginning of the Materials and Methods section. To address the selection of the isolates (which is possibly why a flow chart is requested) we have also added ‘Representative isolates that were PCR positive for rmpA/rmpA2, belonged to high-risk clones (identified by VNTR typing) and in which a carbapenemase gene(s) had been detected were selected for this study’ following the description of these methods (line 58 in the original version, lines 60-62 in the revision). It just seems simpler to address this like that than by a flow-chart. We hope that answers the request. Please note also that we say ‘Fortunately, the incidence of high-risk clones with virulence plasmids was still relatively low even among submissions to the reference laboratory, which are enhanced both for carbapenemase producers and for potentially hypervirulent isolates; we have noted them since 2016 and have found a total of 40 non-duplicate isolates among some 3,600 isolates screened up until the end of the first quarter of 2019’ (lines 261 – 265) to give an idea of the incidence of such isolates among referrals to the UK reference laboratory.